# Third Molar Eruption in Dental Panoramic Radiographs as a Feature for Forensic Age Assessment—Presentation of a New Non-Staging Method Based on Measurements

**DOI:** 10.3390/biology12111403

**Published:** 2023-11-05

**Authors:** Maximilian Timme, Jostin Bender, Laurin Steffens, Denys Shay, Andreas Schmeling

**Affiliations:** 1Institute of Legal Medicine, University Hospital Münster, Röntgenstraße 23, 48149 Münster, Germany; j_bender@posteo.de (J.B.); laurin.steffens@gmx.de (L.S.); denys_shay@g.harvard.edu (D.S.); andreas.schmeling@ukmuenster.de (A.S.); 2Department of Epidemiology, Harvard T.H. Chan School of Public Health, 677 Huntington Ave, Boston, MA 02115, USA

**Keywords:** age estimation, forensic dentistry, dental age, orthopantomogram, wisdom tooth

## Abstract

**Simple Summary:**

The aim of the study was to develop a method for assessing the eruption of mandibular wisdom teeth based on measurements, rather than using fixed stages within the framework of the forensic age assessment. A total of 423 dental panoramic radiographs (DPRs) from individuals aged 15 to 25 were examined, and measurements were taken by two independent examiners. It was possible to develop a method comparing the distances between certain dental landmarks and a simplified radiological occlusal plane. The results revealed moderate correlations between these measurements and individuals’ chronological ages. Intra- and inter-rater reliability for the new method were excellent. In conclusion, this study proposes a novel and promising method for assessing the eruption status of mandibular wisdom teeth in forensic age assessment using DPRs. Further research is needed to validate these findings through reference studies.

**Abstract:**

The evaluation of third molar eruption in dental panoramic radiographs (DPRs) constitutes an evidence-based approach for forensic age assessment in living individuals. Existing methodologies involve staging morphological radiographic findings and comparing them to reference populations. Conversely, the existing literature presents an alternative method where the distance between third molars and the occlusal plane is measured on dental plaster models. The aim of this study was to adapt this measurement principle for DPRs and to determine correlation between eruption and chronological age. A total of 423 DPRs, encompassing 220 females and 203 males aged 15 to 25 years, were examined, including teeth 38 [FDI] and 48. Two independent examiners conducted the measurements, with one examiner providing dual assessments. Ultimately, a quotient was derived by comparing orthogonal distances from the mesial cementoenamel junctions of the second and third molars to a simplified radiological occlusal plane. This quotient was subsequently correlated with the individual’s age. We estimated correlations between age and quotients, as well as inter- and intra-rater reliability. Correlation coefficients (Spearman’s rho) between measurements and individuals’ ages ranged from 0.555 to 0.597, conditional on sex and tooth. Intra-rater agreement (Krippendorf’s alpha) ranged from 0.932 to 0.991, varying according to the tooth and sex. Inter-rater agreement ranged from 0.984 to 0.992, with distinctions drawn for different teeth and sex. Notably, all observer agreement values fell within the “very good” range. In summary, assessing the distance of third molars from a simplified occlusal plane in DPRs emerges as a new and promising method for evaluating eruption status in forensic age assessment. Subsequent reference studies should validate these findings.

## 1. Introduction

Forensic age assessment has emerged as a focal point in forensic science research [1,2,3,4,5,6,7,8,9,10,11,12]. Within the field of forensic age assessment, dental methods have maintained their significance and continue to be integral components of relevant professional recommendations [13,14,15]. Determining the age of living individuals, whose age is unknown for various reasons or whose age claims are subject to doubt, poses a substantial challenge within the domain of forensic dentistry [16,17,18,19,20].

A dental feature for which the correlation with age is well substantiated by good evidence is the eruption of the mandibular third molars in the dental panoramic radiograph (DPR) [21,22,23,24,25,26,27,28]. A major advantage of eruption as an age assessment feature is its predominantly genetic determination and its substantial independence from external factors [29,30,31]. Various methods exist for assessing third molar eruption in DPR, and a commonality across all these methods is the description of eruption using defined morphological stages, corresponding to an ordinal scale [21,32,33,34,35]. Thus, in these methods, eruption is conceptualized as a categorical variable. Previously, our group demonstrated that among the existing staging classifications, the 2012 classification proposed by Olze et al. is particularly well suited for future studies [36].

Within the forensic science community, there has been a long-standing debate among experts regarding the suitability of methods that rely on the assignment of fixed morphologic stages for age assessment [37]. While numerous studies have substantiated the suitability of individual methods [21,22,24,26,36], others argue that reliance on fixed morphologic stages has inherent limitations and is subjective [37,38]. For example, it has been proposed that when assigning specific developmental stages, intermediate phases may sometimes be neglected, creating the impression of a sudden shift from one stage to a higher one. To illustrate, an individual may remain categorized within a particular stage for an extended period and then progress to a higher stage from one day to the next. Critics highlight this aspect as a significant shortcoming of staging methods [37]. Some propose capturing age estimation characteristics through metric variables, described using parametric methods [37,38,39,40]. This perspective suggests replacing morphological stages with measurements. However, as of now, there is insufficient evidence to conclusively assert the practical superiority or tangible benefits of parametric methods.

In 2005, Wedl and Friedrich introduced a method for eruption assessment based on measurements [41]. The authors measured the distance of the third molars from the occlusal plane in both the upper and lower jaw. Specifically, dental plaster models were articulated, and interocclusal records of the third molar regions against the sagittal occlusal curve (Spee-curve) were obtained. These records were subsequently meticulously measured to the nearest tenth of a millimeter using a caliper. In addition, each model had to be accompanied by a corresponding DPR. In the radiographs, an assessment was conducted to determine whether the teeth were appropriately aligned in their axes or if any other impediments to eruption were present. Hence, X-ray imaging was not employed to directly assess the eruption process itself, but rather to ascertain the feasibility of the analyses on the dental plaster models.

It is worth mentioning that Wedl and Friedrich did not identify a relationship between the distance of the third molars from the occlusal plane and age. The scatterplots corresponding to their findings did not exhibit a discernible trend [41]. In summary, it is imperative to acknowledge that this method, which necessitates the availability of dental models and radiographic images, presents a formidable challenge. Furthermore, the measurements conducted within this framework exhibit a high degree of intricacy. Nonetheless, the pursuit of a parametric methodology for evaluating the eruption of third molars persists.

The objective of the present study was to develop a measurement-based method for assessing the eruption of mandibular third molars in the DPR and to examine its correlation with age and its reliability.

## 2. Materials and Method

The dental panoramic radiographs, which were all digital, were obtained from a university dental clinic (Münster, North Rhine-Westphalia, Germany).

For this study, the age range was selected to encompass the eruption of mandibular third molars as comprehensively as possible. In line with the existing literature on the subject, an age range of 15 to 25 years was chosen [36]. Furthermore, the aim was to achieve a balanced distribution between sexes. The overall objective was to gather a total of 25 radiographic images per age group and sex.

The inclusion criteria for this study required sufficient image quality to facilitate the radiological detection of teeth 38 (FDI) and 48. First, sufficient sharpness of the images was checked visually. Then, positioning errors of the examined person in the X-ray device were elicited by visual assessment. For this assessment, the scientifically validated rules of thumb were used [42,43].

Additionally, individuals with displaced or retained third molars were excluded. Retentions were assessed based on established clinical criteria, with an exclusion criterion of an angle exceeding 30 degrees in the mesial direction [44,45]. Disto-angulation always resulted in exclusion, regardless of the degree. Cases in which lesions or dental treatments on the first and second molar made it impossible to assign the simplified occlusal plane were also excluded. Furthermore, individuals displaying evidence of genetic eruption disorders or other impediments to eruption were excluded.

The dental clinic encompassed various specialized departments. Consequently, the study cohort comprised a convenience sample of individuals seeking treatment across the domains of dental surgery, orthodontics, prosthodontics, and conservative dentistry [46,47]. Furthermore, the radiographic images, originally captured for clinical purposes, underwent retrospective assessment in alignment with the current research inquiry.

It was imperative that the precise ages of the participants at the time of their X-ray examinations were well documented. The radiographs were scrutinized in DICOM format, employing suitable workstations equipped with synedra Personal View software version 22.0.0.2 (developed by synedra information technologies GmbH, Innsbruck, Austria). The assessment process routinely made use of the software’s magnification and gray-level adjustment tools. The examining panel comprised two board-certified dentists. One of the two dentists performed two independent evaluations. There were 6 weeks between the evaluations.

The methodology was determined during a first separate phase of the study. Within the scope of this preliminary investigation, various approaches were compared using 50 DPRs from the subsequent study sample. For example, different reference points were compared, different variants of the occlusal plane and different axes of the connecting lines. The correlation with age and the reliabilities were also determined during this preliminary examination. The procedure published here proved to be the most practical and accurate among them.

In the process of measurement, the following steps were performed:

Initially, a zoom-in procedure was applied to focus on the specific region of interest, ensuring that the mandibular teeth from the first to the third molars were fully visualized. Subsequently, the simplified occlusal plane was delineated within the image. This plane was defined as extending from the radiologically highest mesial point of the first molar to the radiologically highest distal point of the second molar. From the simplified occlusal plane, two perpendicular lines were then introduced. The first perpendicular line extended from the occlusal plane to the radiological mesial cementoenamel junction (CEJ) of the second molar. The second perpendicular line extended from the occlusal plane to the mesial CEJ of the third molar. The length of these two perpendicular lines is measured, and a quotient is calculated. The length is recorded in millimeters with one decimal place where they were used as dimensionless numbers for a quotient. The quotient is then correlated with the individual’s age as a measure for age assessment (Figure 1 and Figure 2).

Spearman’s rank correlation coefficient evaluated the correlation between age and quotient [48]. Krippendorff’s alpha was used to evaluate agreement between and within raters [49,50]. Repeatability for one rater and the reproducibility of the two raters were investigated as means of evaluating the reliability of the method. Confidence intervals were calculated at the 95% level.

## 3. Results

For the present study, a total of 423 out of the 550 originally approached digital DPRs from 220 females and 203 males aged 15.04 to 25.98 years were included in the study. Table 1 shows the composition of the study cohort by age and sex.

The main reason for the exclusion of radiographs was a displacement of the third molars of more than 30 degrees and the absence or pathological changes on the first and second molars, which made it impossible to define the simplified occlusal plane. Additionally, in multiple radiographs, the lower third molar was missing or affected by pathologies.

The coefficients of the correlation between quotient and age ranged between ρ = 0.555 and ρ = 0.597, depending on the tooth and sex (Table 1). Based on these values, the correlations for tooth 38 were slightly lower for both sexes compared to tooth 48. However, given the almost entirely overlapping 95% confidence intervals for both sexes and both teeth, this observation appeared to be statistically non-significant at an alpha of 0.05 (Table 1). All values fall within the realm of moderate correlation, defined as a correlation coefficient range between 0.41 and 0.60 [48]. However, when examining the upper limit of the 95% confidence intervals, these values also encompass the domain of strong correlation, typically established between 0.61 and 0.80 [48].

Intra-rater agreement values ranged from α = 0.932 to α = 0.991 (Table 2). Notably, among males, tooth 48 exhibited slightly lower values compared to the other assessments.

As all values exceeded 0.8, according to Krippendorff, one can infer a high level of agreement [49,50]. According to Landis and Koch, values above 0.81 are classified as “almost perfect” [51].

Inter-rater agreement values ranged from α = 0.984 to α = 0.992 (Table 3), falling within a similar range as the intra-rater agreement values (Table 4). They can also be categorized within the “almost perfect” range according to Landis and Koch [51].

## 4. Discussion

The present study introduces a measurement-based method for assessing the eruption of mandibular third molars in DPR, as opposed to stage-based approaches. Principally, the study delves into the correlation between eruption and chronological age. The primary aim of this study was not to proffer regression formulas suitable for immediate utility, nor to provide practically actionable reference values.

Devising the actual methodology presented a unique challenge and was formulated following a preliminary investigation that involved the comparison of various approaches using 50 DPRs from the present sample.

For our feasibility study of a new methodology, all distoangulated teeth were first excluded without consideration of the degree of displacement. This decision was based on the assumption that the distoangulation in some ways contributes to impaction, because the tooth tends to go in the direction towards to bone. A possible bias should be excluded here. However, a subsequent study should examine the influence of distoangulation of the teeth on the method more specifically.

The development of the method commenced with the inquiry into the optimal means of transferring the occlusal plane onto the radiographic image. Ultimately, the decision was made to employ a simplified occlusal plane that could be generated within a minimized, magnified image frame. This approach was designed to enable work within the selected and magnified image frame and thus substantially streamline the process. In addition, distortions due to possible projection errors should be kept as low as possible. Consequently, the first and second molars were designated as reference points. Since specific cusps in the DPR are challenging to delineate, clearer reference points were established to define the simplified occlusal plane. The radiologically highest cuspal points, mesially on the first molar and distally on the second molar, were found to be unequivocal reference points, irrespective of whether they were buccal or lingual cusps. The emphasis was on the clarity of the point.

Next, the measurement points on the teeth needed to be established. This consideration stems from the fact that the DPR is not isometric and distortions regularly occur in the DPR, especially in the molar region [52,53,54,55,56,57]. This means that the measurement of absolute numbers in the DPR should be dispensed with to the extent possible. Therefore, a quotient should be calculated in each instance—in contrast to the simple measurement of the distance of the third molar from the occlusal plane. In addition, it was important for the individual lengths in the quotient to be as close as possible in the horizontal plane, given the various distortions present in the DPR. Consequently, measurement points should be defined on the second and third molars. Using a cusp tip was ruled out, as by definition, for the second molar, it lies on, or at least very close to, the occlusal plane, which would make the measurement exceedingly challenging and yield a value close to zero. The mesial CEJ of both teeth were found to be remarkably distinct and practical for implementation. It is important to note that this method requires the CEJ to be clearly discernible for its application. This restricts the method’s applicability to adequately mineralized teeth. In practice, this aspect did not pose a limitation within the examined age groups.

We then considered the angle for measuring the distance between the reference points and the simplified occlusal plane. Initially, the aim was to align with the axis of each tooth. However, in practice, this approach proved to be impractical, as the preliminary investigations revealed substantial discrepancies in inter-observer agreement. Therefore, a compromise was adopted, significantly simplifying the procedure. It was determined that the orthogonal line to the occlusal plane passing through the reference point should be utilized. This simplification resulted in a significant reduction in potential errors.

The excellent inter-observer agreement values obtained in the current study demonstrate the high reproducibility of this methodology.

In comparison to the method introduced by Wedl and Friedrich in 2005, significant improvements have been achieved with the current approach [41]. Unlike the Wedl and Friedrich method, which necessitated the creation of dental plaster models and bite registration, our method eliminates these requirements. Additionally, it is noteworthy that while the Wedl and Friedrich method utilized a DPR to identify correct tooth axes and potential impactions, it ultimately assessed only those teeth that had already erupted through the gingiva. In contrast, our proposed method allows for the evaluation of teeth still completely encased within the bone.

Furthermore, Wedl and Friedrich did not find a correlation between the distance of third molars from the occlusal plane and chronological age [41]. This discrepancy may arise from differences in the studied cohorts or potential limitations in their methodology.

This study, as measured by Spearman’s rho, revealed only a moderate correlation between the calculated quotient and chronological age [48]. Naturally, it is imperative to consider whether this is indicative of a methodological limitation. Thus, the DPR is subject to its inherent distortion, especially in the molar region [52,53,54,55,56,57], which affects any radiological method of evaluating the eruption using DPR to a certain extent. These effects would not occur with methods that do not require DPR or use an isometric X-ray method. Here, some studies have been published in the past using cone-beam computed tomography (CBCT) for age assessment [58,59,60,61]. This method is isometric and would thus solve the problem of distortions. However, this approach can be criticized for its greater radiation dispersion than DPR. In addition, studies on the use of magnetic resonance imaging (MRI) for age assessment on teeth have also been published in the past [24,62,63]. Here, the radiation disposition is completely omitted. Promising results were also obtained for the assessment of third molar eruption on MRI for age assessment [24]. Whether these results can be transferred the method presented in the present study should be verified in the future-whereby, of course, it must be stated that the calculation of a quotient could be dispensed with in the case of isometric methods, since it is legitimate in such isometric methods to measure distances directly. However, the considerably higher costs, the effort required for the examination and the significantly more limited availability must be mentioned as disadvantages for the use of MRI in age assessment.

In addition, it must be assumed that minor forms of angulation, which were below the exclusion limit, could nevertheless have influenced the results. Moreover, this problem is also independent of imaging. In the future, further studies must clarify what influence such minor forms actually have.

However, it is crucial to note that these results for correlation between age and eruption align with the findings reported in the relevant literature. Our group previously reported Spearman’s rho values of 0.583 (males) and 0.445 (females) for the Olze et al. Method [36]. Conversely, Olze et al. themselves indicated η-coefficients ranging from 0.610 to 0.699 for their method [21]. Therefore, it’s worth noting that other research groups have identified correlations categorized as “Moderate” and “Strong“ [48]. However, it is prudent to exercise caution when attributing the differences between our results and those from the literature, as these disparities are likely influenced by sample heterogeneity.

Overall, determination of the eruption of mandibular third molars in the DPR is a fast, cost-effective and straightforward method for forensic age assessment. The limitation arises from the only moderate correlation between feature and age. Potentially, the correlation could be further improved in the future by even more specific inclusion or exclusion criteria. This could include, for example, the question of how to deal with minor forms of angulation.

Nonetheless, it can be tentatively concluded that the introduction of the measurement-based method presented here does not lead to a substantial improvement in the correlation between the characteristic and chronological age. Considering the call from some forensic age assessment experts to replace morphology-based stage classifications with measurement-based methods [37,38], it can be assumed that the limitations of the characteristics for use in forensic age assessment ultimately stem from the inherent variability of the characteristic itself. Regardless of the evaluation method employed, each characteristic will ultimately retain its statistically acknowledged level of imprecision.

In light of this perspective, one must consider the substantial additional elaboration required when applying the present method compared to a stage-based approach that does not involve measurements. The authors justify this increased effort not primarily based on the correlation with chronological age but rather due to the improved inter-rater reliability it offers.

In comparison to data from the literature regarding stage-based methods on DPR, the present inter-rater agreement yields significantly better results. Our group explored various stage-based approaches within a cohort of 211 DPRs from individuals aged 15 to 25 years [36]. Their findings favored the method proposed by Olze et al. in 2012 as the most suitable. Our groups previously reported Krippendorff’s alpha values for inter-rater agreement of 0.797 (males) and 0.792 (females) for the Olze et al. method, which outperformed the other methods they investigated, all of which yielded inferior results [36]. It is worth noting, however, that our group conducted their investigations with three different examiners, making direct comparisons challenging [36]. Other authors report a Cohens kappa of 0.76 for inter-rater reliability for the Olze et al. method [64]. For the visual clinical assessment of intraoral eruption, values as high as 100% for inter-rater reliability have been published [65].

Nevertheless, it can be reasonably assumed that the present measurement-based method benefits from clear reference points for measurement. Furthermore, the inherent ambiguity in stage-based methods, particularly in borderline cases, is obviated by this approach, as no staging decisions are necessitated.

Looking ahead to the future of forensic age assessment, it becomes evident that the path leads toward fully automated procedures reliant on artificial intelligence [66,67,68,69,70,71,72,73,74,75]. This prompts a broader question about the relevance of the present research. On the one hand, the journey towards automated age assessment is a long one, and some proposed methods involve even greater complexity and resource demands than established ones. Reliable methods will continue to be necessary until the full establishment of automated and practicable procedures.

On the other hand, studies such as the one at hand contribute to a fundamental understanding of the characteristics and principles underlying age assessment as a process. The authors contend that this understanding will remain essential in evaluating the results of automated software, even when fully automated methods become prevalent. From the authors’ perspective, research like the present study thus plays a crucial and irreplaceable role in advancing foundational knowledge in this field.

## 5. Conclusions

The measurement-based method presented in this study for assessing mandibular third molar eruption in DPR shows potential for future research applications, demonstrating high reliability.

Although the correlation between eruption and chronological age was moderate, this finding is consistent with the previous literature. Such moderate correlation, across different studies, suggests that the imprecision stems from the inherent variability of the eruption process itself, rather than the analysis method used.

## Figures and Tables

**Figure 1 biology-12-01403-f001:**
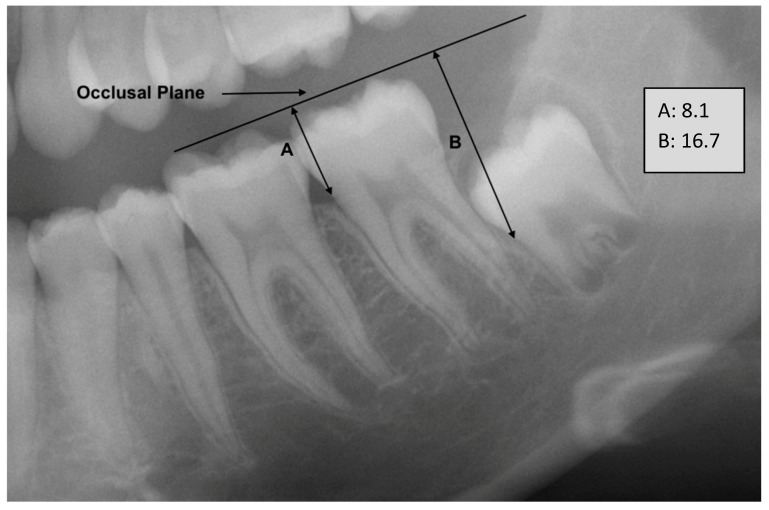
Example figure from the cohort with a little advanced eruption of tooth 38 (FDI) of a 15-year-old female. Lengths A and B given in millimeters. The occlusal plane has simplified representation as described in the method.

**Figure 2 biology-12-01403-f002:**
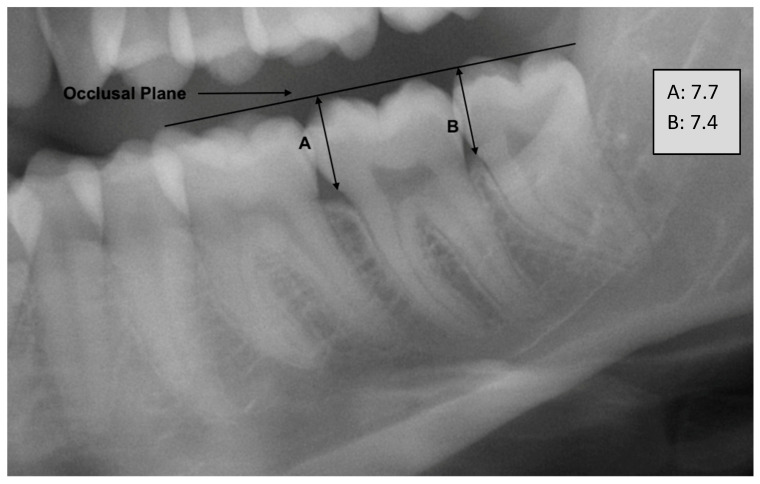
Example figure from the cohort with fully advanced eruption of tooth 38 [FDI] of a 25-year-old female. Lengths in millimeters. The occlusal plane has simplified representation as described in the method.

**Table 1 biology-12-01403-t001:** Age and sex distribution of the sample.

Age	Females (*n*)	Males (*n*)	Total (*n*)
15	21	18	39
16	18	23	41
17	20	21	41
18	22	20	42
19	21	20	41
20	20	17	37
21	20	20	40
22	18	15	33
23	24	21	45
24	16	14	30
25	20	14	34
Total (*n*)	220	203	423

**Table 2 biology-12-01403-t002:** Spearman correlation (ρ) between stage and age, for each tooth and sex. The basis for the data is the second assessment run of one of the examiners.

Tooth	Sex	Spearman ρ	95% LCL	95% UCL
38	Male	0.557	0.441	0.637
48	Male	0.581	0.455	0.671
38	Female	0.555	0.423	0.647
48	Female	0.597	0.476	0.688

LCL: lower confidence limits; UCL: upper confidence limits.

**Table 3 biology-12-01403-t003:** Krippendorff coefficients (*α*) measuring intra-rater repeatability for each tooth by sex.

Tooth	Sex	Krippendorff’s α	95% LCL	95% UCL
38	Male	0.984	0.968	0.99
48	Male	0.932	0.735	0.993
38	Female	0.986	0.977	0.989
48	Female	0.991	0.988	0.993

LCL: lower confidence limits; UCL: upper confidence limits.

**Table 4 biology-12-01403-t004:** Krippendorff coefficients (***α***) measuring inter-rater reliability for each tooth by sex.

Tooth	Sex	Krippendorff’s α	95% LCL	95% UCL
38	Male	0.984	0.971	0.988
48	Male	0.986	0.983	0.99
38	Female	0.992	0.99	0.994
48	Female	0.99	0.986	0.992

LCL: lower confidence limits; UCL: upper confidence limits.

## Data Availability

The datasets generated during the current study are available from the corresponding author on reasonable request.

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
