# Peer review of "Third Molar Eruption in Dental Panoramic Radiographs as a Feature for Forensic Age Assessment—Presentation of a New Non-Staging Method Based on Measurements"

_biology, 2023, doi:10.3390/biology12111403_

Round 1
Reviewer 1 Report
Comments and Suggestions for Authors
Dear authors:
I would like to convey my congratulations for your work. I only have a couple of questions to tell you about it.
On the one hand, a statistical question. The results of the spearman rank are presented in positive values, but if I have understood correctly, when age increases the ratio between the occlusal lengths of second and third molars should decrease and therefore the Spearman rank should be negative. It's right?
On the other hand, in their analysis of the efficiency of the method they point out possible problems derived from the method itself (lines 251 and 252). Although they have already been noted in the introduction, I think it would be good to highlight the importance of the effect that the distortion derived from the radiographic technique induces in the image of the third molars. Likewise, although impacted and angulated third molars have been excluded, it is extremely difficult to exclude the effect of minor angulations in the measurements collected. In any case, the proposed methodology brilliantly automates the taking of measurements and is undoubtedly a notable finding of his work.
Author Response
We the authors thank the reviewers for their valuable work. Our manuscript has benefited greatly from the reviewer's comments. We have implemented all the reviewer's suggested changes and have modified our text in the appropriate places. In the following, we respond to all comments.
Reviewer I
Comment 1:
On the one hand, a statistical question. The results of the spearman rank are presented in positive values, but if I have understood correctly, when age increases the ratio between the occlusal lengths of second and third molars should decrease and therefore the Spearman rank should be negative. It's right?
Response:
For this context, only the choice of numerator and denominator for the quotient is decisive. With the dimensionless quotient, the values could be freely swapped, which is the point of the reviewer's comment. We decided to take the quotient in such a way that it increases with increasing age. This also makes the correlation coefficient positive. This choice also simplifies comparison with data from the literature. In addition, a positive quotient of feature and age is, at first glance, more life-like. Therefore, we consider the choice as made by us to be reasonable.
Comment 2:
On the other hand, in their analysis of the efficiency of the method they point out possible problems derived from the method itself (lines 251 and 252). Although they have already been noted in the introduction, I think it would be good to highlight the importance of the effect that the distortion derived from the radiographic technique induces in the image of the third molars. Likewise, although impacted and angulated third molars have been excluded, it is extremely difficult to exclude the effect of minor angulations in the measurements collected. In any case, the proposed methodology brilliantly automates the taking of measurements and is undoubtedly a notable finding of his work.
Response:
We completely agree with the reviewer's view and have enhanced our text accordingly.
Discussion:
[…] Thus, the DPR is subject to its inherent distortion, especially also in the molar region [50 – 55], which affects any radiological method of evaluating the eruption using DPR. These effects would not occur with methods that do not require DPR or use an isometric X-ray method. In addition, it must be assumed that minor forms of angulation, which were below the exclusion limit, could nevertheless have influenced the results. […]
Closing statement to the editor:
By incorporating the reviewers' comments, the text has been expanded as requested.
We'd like to conclude by expressing our appreciation
to the editor for her work.

Reviewer 2 Report
Comments and Suggestions for Authors
Dear Editor of Biology and Authors,
I have read the paper titled «Third molar eruption in dental panoramic radiographs as a feature for forensic age assessment - Presentation of a new non-staging method based on measuring» with interest and I find it potentially publishable in the Special Issue «Recent Advances in Forensic Anthropology: From Experience to Current Applications and Innovative Approaches». It shows good scholarship as it is well researched and written, and features a noteworthy exploratory analysis of dental eruption in relation with age that has a great relevance for forensic identification. I have some doubts and suggestions that I would like to be addressed by the authors of this paper.
1. The title needs a retouch, the second part is not clear (Presentation of a new non-staging method based on measuring): measuring what?; maybe replace “measuring” by “measurements”?
2. Delete “since antiquity” (p. 1, line 38).
3. Why use Spearman Rank coefficient and not Pearson? Variables are continuous (the quotients of the two measurements are a continuous distributed variable, i.e., they show an infinite number of values within a certain range) and the relation between them is linear. Or this is not the case?
4. The correlation between measurements and age is moderate and that finding, in addition to the results of other studies, suggest that measurements of the third molar eruption may be of use for age estimation BUT the potential for it is not great. This must be clearly stated.
My best regards.
Author Response
We the authors thank the reviewers for their valuable work. Our manuscript has benefited greatly from the reviewer's comments. We have implemented all the reviewer's suggested changes and have modified our text in the appropriate places. In the following, we respond to all comments.
Reviewer II
Comment 1:
The title needs a retouch, the second part is not clear (Presentation of a new non-staging method based on measuring): measuring what?; maybe replace “measuring” by “measurements”?
Response:
We have changed the title as proposed:
Third molar eruption in dental panoramic radiographs as a feature for forensic age assessment - Presentation of a new non-staging method based on measurements.
Comment 2:
Delete “since antiquity” (p. 1, line 38).
Response:
We have changed the text accordingly and removed the aspect.
Comment 3:
Why use Spearman Rank coefficient and not Pearson? Variables are continuous (the quotients of the two measurements are a continuous distributed variable, i.e., they show an infinite number of values within a certain range) and the relation between them is linear. Or this is not the case?
Response:
We have discussed this comment in detail with the statistics expert in our group. He would like to respond to the comment as follows:
Thank you for this excellent question. The assumptions required for a Pearson’s Correlation Coefficient are that both variables are 1) continuous and 2) normally distributed, and 3) that the relationship between them is linear. While 1) and 3) are fulfilled, 2) is not fulfilled because neither age nor the quotient are normally distributed. Therefore, it is more appropriate to use a Spearman’s Rank Correlation Coefficient which does not assume a normal distribution of the two variables.
Comment 4:
The correlation between measurements and age is moderate and that finding, in addition to the results of other studies, suggest that measurements of the third molar eruption may be of use for age estimation BUT the potential for it is not great. This must be clearly stated.
Response:
Thank you for raising this important point. We have adjusted our discussion, stating the limited potential.
[…] Overall, determination of the eruption of mandibular third molars in the DPR is a fast, cost effective and straightforward method for forensic age assessment. The limitation arises from the only moderate correlation between feature and age. Potentially, the correlation could be further improved in the future by even more specific inclusion or exclusion criteria. This could include, for example, the question of how to deal with minor forms of angulation. […]
Closing statement to the editor:
By incorporating the reviewers' comments, the text has been expanded as requested.
We'd like to conclude by expressing our appreciation
to the editor for his work.

Reviewer 3 Report
Comments and Suggestions for Authors
Dear authors,
thank you for the interesting manuscript you submitted and for the excellent work you have done. The manuscript “Third molar eruption in dental panoramic radiographs as a feature for forensic age assessment - presentation of a new non-staging method based on measuring” has the characteristic of highlighting the state of the art of the issue even to an attentive reader and not specialized in the forensic field. The content of the manuscript is communicated clearly and appropriately and addresses a very valid research question. The title provides a clear indication of the focus of the manuscript which is consistent throughout. The abstract is an accurate summary of the objectives, results and conclusions. The introduction clearly summarizes the current state of the topic highlighting pros and cons.
Furthermore, the introduction clearly defines the purpose of the study and is consistent with the rest of the manuscript. The study design and approach have been sufficiently described and appropriately address the research question. The authors have included all essential information necessary to support the findings presented in the manuscript. The purpose of the manuscript has some meaning, since the main purpose is not to provide regression formulas or static reference values. This is the result of a deep understanding of the limitations of forensic chronological age identification. The inclusion and exclusion criteria for selecting the sample were fair and balanced. The tables explain the composition of the study cohort well. The definition of the correlation coefficient was calculated correctly and all the statistical analysis was well conducted.
Here are some tips you can consider to improve your manuscript:
- Lines 56-59: “For instance, it is suggested that in the assignment of specific stages, transitional zones might be overlooked, leading to the appearance of an abrupt transition from one stage to a higher one. For instance, an individual might be assigned a particular stage for years, only to advance to a higher stage from one day to the next..” I suggest rephrasing these sentences.
- Line 102: “Distoangulation always resulted in exclusion, regardless of the degree” You should add why you excluded teeth with any degree of distoangulation.
- Line 120: “The methodology was ascertained through an initial facet of the investigation”. Please rephrase.
- Line 206: You could explain to readers the criteria you used to identify the DPR that was good enough for the study. How to evaluate its validity?
- Finally, given the discursive nature of the discussion, some suggestions for future studies and perspectives will be appreciated. For example:
o How to overcome limitations such as displaced teeth or diseases?
o What about chronological age assessment on 3D radiographs when they are available?
For the reasons above, I suggest evaluating your work for publication after minor revision.
Best regards
Comments on the Quality of English Language
Minor editing of English language required.
Author Response
We the authors thank the reviewers for their valuable work. Our manuscript has benefited greatly from the reviewer's comments. We have implemented all the reviewer's suggested changes and have modified our text in the appropriate places. In the following, we respond to all comments.
Reviewer III
Comment 1:
Lines 56-59: “For instance, it is suggested that in the assignment of specific stages, transitional zones might be overlooked, leading to the appearance of an abrupt transition from one stage to a higher one. For instance, an individual might be assigned a particular stage for years, only to advance to a higher stage from one day to the next..” I suggest rephrasing these sentences.
Response:
We have revised the passage as follows:
For example, it has been proposed that when assigning specific developmental stages, intermediate phases may sometimes be neglected, creating the impression of a sudden shift from one stage to a higher one. To illustrate, an individual may remain categorized within a particular stage for an extended period and then progress to a higher stage from one day to the next.
Comment 2:
Line 102: “Distoangulation always resulted in exclusion, regardless of the degree” You should add why you excluded teeth with any degree of distoangulation.
Response:
We have added the following aspect to our text in the Discussion:
[…] For our feasibility study of a new methodology, all disto-angulated teeth were first excluded without consideration of the degree of displacement. This decision was based on the assumption that the distoangulated angulation in some ways contributes to impaction, because the tooth tends to go in the direction towards to bone. A possible bias should be excluded here. However, a subsequent study should examine the influence of distoangulation of the teeth on the method more specifically. […]
Comment 3:
Line 120: “The methodology was ascertained through an initial facet of the investigation”. Please rephrase.
Response:
We have revised the passage as follows:
The methodology was determined during a first separate phase of the study.
Comment 4:
Line 206: You could explain to readers the criteria you used to identify the DPR that was good enough for the study. How to evaluate its validity?
Response:
We have added the following aspect to our text in the Material and Method section:
First, sufficient sharpness of the images was checked visually. Then, positioning errors of the examined person in the X-ray device were elicited by visual assessment. For this assessment, the scientifically validated rules of thumb were used [42, 43].
Comment 5:
Finally, given the discursive nature of the discussion, some suggestions for future studies and perspectives will be appreciated. For example:
o How to overcome limitations such as displaced teeth or diseases?
o What about chronological age assessment on 3D radiographs when they are available?
Response:
We have added the following aspect to our text in the Discussion:
[…] Thus, the DPR is subject to its inherent distortion, especially also in the molar region [52 – 57], which affects any radiological method of evaluating the eruption using DPR to a certain extent. These effects would not occur with methods that do not require DPR or use an isometric X-ray method. Here, some studies have been published in the past using cone-beam computed tomography (CBCT) for age assessment [58–61]. This method is isometric and would thus solve the problem of distortions. However, this approach can be criticized for its greater radiation dispersion than DPR. In addition, studies on the use of magnetic resonance imaging (MRI) for age assessment on teeth have also been published in the past [24, 62, 63]. Here, the radiation dis-position is completely omitted. Promising results were also obtained for the assessment of third molar eruption on MRI for age assessment [24]. Whether these results can be transferred the method presented in the present study should be verified in the future - whereby, of course, it must be stated that the calculation of a quotient could be dispensed with in the case of isometric methods, since it is legitimate in such isometric methods to measure distances directly. However, the considerably higher costs, the effort required for the examination and the significantly more limited availability must be mentioned as disadvantages for the use of MRI in age assessment.
In addition, it must be assumed that minor forms of angulation, which were below the exclusion limit, could nevertheless have influenced the results. Moreover, this problem is also independent of imaging. In the future, further studies must clarify what influence such minor forms actually have. […]
Closing statement to the editor:
By incorporating the reviewers' comments, the text has been expanded as requested.
We'd like to conclude by expressing our appreciation
to the editor for his work.
